# Open Renal Transplantation in Obese Patients: A Correlation Study between BMI and Early and Late Complications with Implementation of a Prognostic Risk Score

**DOI:** 10.3390/life14070915

**Published:** 2024-07-22

**Authors:** Sara Marzorati, Domenico Iovino, Davide Inversini, Valentina Iori, Cristiano Parise, Federica Masci, Linda Liepa, Mauro Oltolina, Elia Zani, Caterina Franchi, Marika Morabito, Mattia Gritti, Caterina Di Bella, Silvia Bisogno, Alberto Mangano, Matteo Tozzi, Giulio Carcano, Giuseppe Ietto

**Affiliations:** 1General, Emergency and Transplant Surgery Department, ASST-Sette Laghi, 21100 Varese, Italy; saramarzorati20@gmail.com (S.M.); giulio.carcano@uninsubria.it (G.C.); 2Department of Medicine and Innovation Technology (DiMIT), University of Insubria, 21100 Varese, Italy; 3Department of General Surgery, Humanitas Clinical and Research Center, 20089 Rozzano, Italy; mattia.gritti95@gmail.com; 4Kidney and Pancreas Transplantation Unit, Department of Surgery, Oncology and Gastroenterology, Padova University Hospital, 35128 Padova, Italy; caterina.dibella.1@unipd.it; 5Department of Cardiologic Intensive Care, Hemodynamics and Cardiology, S.M. Goretti Hospital, Sapienza University of Rome, 04100 Latina, Italy; s.bisogno@ausl.latina.it; 6Division of General, Minimally Invasive and Robotic Surgery, Department of Surgery, University of Illinois at Chicago, Chicago, IL 60607, USA; alberto.mangano@gmail.com; 7Vascular Surgery Department, ASST-Sette Laghi, 21100 Varese, Italy; matteo.tozzi@uninsubria.it; 8Department of Medicine and Surgery (DMC), University of Insubria, 21100 Varese, Italy

**Keywords:** obesity, kidney transplant, prognostic risk score

## Abstract

Background: Obesity is a global epidemic that affects millions worldwide and can be a deterrent to surgical procedures in the population waiting for kidney transplantation. However, the literature on the topic is controversial. This study evaluates the impact of body mass index (BMI) on complications after renal transplantation, and identifies factors associated with major complications to develop a prognostic risk score. Methods: A correlation analysis between BMI and early and late complications was first performed, followed by a univariate and multivariate logistic regression analysis. The 302 included patients were divided into obese (BMI ≥ 30 kg/m^2^) and non-obese (BMI ≤ 30 kg/m^2^) groups. Correlation analysis showed that delayed graft function (DGF) was the only obesity-associated complication (*p* = 0.044). Logistic regression analysis identified female sex, age ≥ 57 years, BMI ≥ 25 and ≥30 kg/m^2^, previous abdominal and/or urinary system surgery, and Charlson morbidity Score ≥ 3 as risk factors for significant complications. Based on the analyzed data, we developed a nomogram and a prognostic risk score. Results: The model’s area (AUC) was 0.6457 (95% IC: 0.57; 0.72). The percentage of cases correctly identified by this model retrospectively applied to the entire cohort was 73.61%. Conclusions: A high BMI seems to be associated with an increased risk of DGF, but it does not appear to be a risk factor for other complications. Using an easy-to-use model, identification, and stratification of individualized risk factors could help to identify the need for interventions and, thus, improve patient eligibility and transplant outcomes. This could also contribute to maintaining an approach with high ethical standards.

## 1. Introduction

Obesity has become a major health issue in developed countries, and its prevalence is steadily increasing globally. According to the latest World Health Organization report, in 2016, obesity (i.e., body mass index (BMI) ≥ 30 kg/m^2^) affected over 650 million adults worldwide, which is approximately 13% of the world’s adult population [1]. On average, according to self-reported data across EU countries, in 2018, more than 1 in 6 adults (17%) were obese, with a rate increase from 11% in 2000 [2]. In the USA, data from NHANES show an obesity prevalence of 41.9% between 2017 and 2020, with an increase from 30.5% in the late 2000s [3]. This global phenomenon is reflected in the population waiting for kidney transplantation, where the percentage of obese kidney recipients has been doubling every 15 years [4,5,6]. In kidney failure, transplantation is recognized as the best long-term therapeutic option in terms of quality of life and life expectancy [7,8]. Nevertheless, obesity, for a variety of reasons, may be a potential contraindication to kidney transplantation. First, there is a chronic organ shortage, such that many more patients are listed for transplantation than those who receive an organ per year. This critical situation overlaps with the debate regarding the actual eligibility of obese patients as candidates for surgery [9]. This is the reason why—faced with the dilemma of whether to deny transplantation to an obese patient—many authors and centres have started to consider a “utility-based” approach in an attempt to allocate the scant organ resources available to the most suitable recipients [10]. Currently, high body mass index (BMI) is a critical consideration when selecting candidates for surgery. However, many authors have questioned the accuracy of BMI as a predictive value in patients on dialysis and waiting for a transplant, as BMI is considered a surrogate measure with significant limitations [11,12]. Moreover, international guidelines and consensus regarding BMI thresholds for transplantation are lacking. According to an international study by Glicklich et al. [13], 30% of centres had no BMI cut-off, 29% used a cut-off level of a BMI of 35 kg/m^2^, and 27% used a BMI of 30 kg/m^2^ as a cut-off. Hence, allocating the limited available organ resources is left to the policies and judgement of single transplant centres. The main deterrent for kidney transplantation in obese patients is the increased risk of perioperative and postoperative complications. Several recent meta-analyses conducted by Nicoletto et al. [14], La Franca et al. [15], Hill et al. [16], and Sood et al. [17] illustrate this situation, which remains controversial, especially regarding graft and patient outcomes. All of these studies pertain to North American patients, whereas the European literature, except for the recent French study by Foucher et al. [18], is still evolving.

One major concern regards the increase in surgical site infections [19]. To reduce infections, the University of Illinois Hospital developed a robotic kidney transplantation method for obese recipients. Robotic surgery proved to be a solution to enable obese patients with kidney failure to access kidney transplantation by reducing the incidence of surgical site infections after kidney transplantation [20]. The Chicago experience is huge, reporting over 200 robotic kidney transplants over 10 years, confirming that robotic surgery is a safe approach for obese patients even in transplant surgery, and also that it guarantees minimal surgical site infection risk [21].

In this context, the present study’s first aim was to analyze the relationship between BMI and the incidence of early and late complications after renal transplantation in an Italian cohort at a single Transplant Centre. Based on the achieved results, the second endpoint was the development of a model to implement a prognostic risk score for the incidence of major complications.

## 2. Materials and Methods

This is a retrospective study with a prospective validation of a collected database of patients who underwent open kidney transplantation between January 2014 and April 2021 at the Hospital “Ospedale di Circolo and Fondazione Macchi” in Varese, Italy. The local institutional review board (IRB) approved the study using protocol n. 0119180 of 19 October 2023. Kidney transplants from both deceased and living donors were included. The cohort comprised 302 patients. The following items were evaluated: the recipients’ demographic characteristics, comorbidities, data regarding the transplantation procedure, length of hospital stay, and the peri- and post-operative complications. Moreover, interdisciplinary nephrological and surgical accurate follow-ups were performed from the day of surgical intervention to the current year following an established internal protocol: every week for the first month, then once a month in the first three months, and finally every six months. The evaluated complications were the same as the recent meta-analyses examined [14,15,16,17,18]. A comparison pertaining to the association between BMI and open renal transplant complications has been performed between the literature data and the results obtained from the present cohort.

All kidney transplants were performed with a standard open technique: through a skin “hockey stick” incision, renal grafts were anastomosed to common-iliac vessels and the bladder with a Lich-Gregoir ureteral reimplantation.

All procedures were performed by senior surgeons (with more than 100 procedures performed), or young surgeons (with more than 30 procedures performed) supervised by a senior. They belonged to the same team, used the same surgical technique, and had similar expertise, according to their different stages of career.

Major and more frequent complications were reported and classified according to the ones in the available literature [9,22,23]. Complications of kidney transplants were classified as early- or late-onset. Early complications included delayed graft function (DGF), acute rejection, wound dehiscence, lymphocele, perirenal hematoma, and incisional hernias. Late complications identified during the follow-up period included new-onset diabetes (NODAT), new-onset arterial hypertension, cytomegalovirus (CMV) infection, neoplasms, and additional hospitalizations due to the surgical procedure and/or immunosuppressive therapy.

The duration of the surgical operation (expressed in minutes) and length of hospitalization (expressed in days) were also evaluated. For the development of the prognostic risk score, the prominent early complications considered (potentially responsible for graft loss) were DGF and acute rejection. The following potential risk factors for such complications were partially identified based on the scientific literature: age, sex, BMI, hypertension, dyslipidemia, diabetes mellitus, and cardiovascular diseases. Other potential risk factors evaluated were previous abdominal and/or urinary system surgery and the Charlson Comorbidity Score. This study corrected age and Charlson Comorbidity Score for higher performance. The age cut-off was set at ≥57 years (similar to the literature, which suggests that it should be set at ≥60 years in surgical predictive scores). This choice was made because the present risk score was designed to be used while evaluating the patient for inclusion in the waiting list, and, according to the latest data from the Italian National Transplant Centre and the National Institute of Health, the average time on the waiting list for a kidney transplant is 3.4 years [24]. During the computation of the Charlson Comorbidity Score, the points related to age—already considered as independent variables—and those related to chronic kidney disease were not factored in; this was performed to obtain a clear and uninfluenced assessment of the effect of every kidney transplant candidate’s comorbidities.

Statistical analysis was performed using the STATA vers.13 software (StataCorp Release 13. Software) (College Station, TX, USA). The sample was analyzed using the following descriptive statistical techniques: mean, standard deviation, minimum, and maximum for the quantitative variables, and absolute and relative frequencies for the categorical variables. The association between variables was first evaluated using the chi-square test for categorical variables and Student’s *t*-test for continuous variables to assess the correlation between BMI and early and late complications, as well as the length of hospital stay and surgical intervention. The significance level was set at *p* < 0.05. Univariate and multivariate logistic regression analyses were performed to identify the risk factors for significant complications. The multivariate logistic regression analysis included variables with an odds ratio (OR) of >1 in the univariate analysis. All variables that remained significant in the multivariate model were used to build a scoring system. A nomogram was implemented to obtain a score and to estimate the probability of developing major complications. To evaluate the effectiveness of the model, the Receiver Operating Characteristic (ROC) curves and performance indicators of the multivariate model were calculated. The area under the curve (AUC), referred to as the absolute value and confidence interval at 95% (95% IC), was also calculated.

## 3. Results

### 3.1. Clinical-Pathological Characteristics of the Cohort

The qualitative and quantitative characteristics of the samples are presented in Table 1 and Table 2, respectively. Among the 302 patients included in the study, 94 were women (31.1%) and 208 were men (68.9%). The number of procedures described for each year is detailed in Table 1. Among the 302 patients, 31 were obese (BMI ≥ 30 kg/m^2^) (10.0%), of whom 5 were class II obese (BMI of 35 kg/m^2^ to <40 kg/m^2^) and 101 were overweight (33.4%) (BMI of 25 kg/m^2^ to <30 kg/m^2^). In total, 62 (20.5%) patients had a Charlson Comorbidity Score ≥ 3; 240 (79.5%) had a score < 3. Five postoperative deaths (1.7%) were recorded within the first three years after transplantation. In total, 236 patients (78.1%) had arterial hypertension in their past medical history, 35 (11.6%) suffered from dyslipidemia, 46 (15.2%) from diabetes mellitus, 119 (39.4%) from cardiovascular disease, and 186 (61.6%) previously underwent abdominal and/or urinary system surgery. The average age at the time of transplant was 53.2 ± 11.67 years (minimum 19 years, maximum 76 years). The average BMI was 24.1 ± 5.5 kg/m^2^ (minimum 15.1 kg/m^2^, maximum 37.7 kg/m^2^). The average Charlson Comorbidity Score was 1.3 ± 1.5 points (minimum 0 points, maximum 7 points). The mean length of hospital stays (LOS) was 22.0 ± 17.7 days (minimum, 6 days; maximum, 224 days), and the mean length of operative time (OT) was 241.0 ± 75.3 min (minimum 110 min, maximum 924 min).

### 3.2. Primary Endpoint: Assessing the Correlation between BMI and Early/Late Complications

The study population was divided into two groups: obese, and non-obese. The association between obesity and incidence of renal transplantation complications was evaluated. Among the early complications, DGF was the most common in both groups. However, this percentage was significantly higher among obese patients (32.3%) compared to non-obese patients (17.3%) with a *p* = 0.044. The occurrence of the other early complications was lower than 10%, and it was very similar in both arms. No statistically significant differences were observed between the two groups (Table 3, Figure 1). Late complications were more frequent, with an incidence higher than 10% in both arms, except for new onset arterial hypertension. More than 40% of patients, in both obese and non-obese cohorts, needed additional hospitalizations due to the complications related to kidney transplant and/or immunosuppressive therapy; over 20% were treated for CMV infection; and more than 10% developed New-onset Diabetes (NODAT) and neoplasms. Obese patients had a higher incidence of NODAT and neoplasms. However, this association with obesity was not statistically significant. The incidence of any late evaluated complications was significantly different among the two groups. (Table 4, Figure 2). Additionally, a correlation analysis between BMI and the LOS (expressed in days) and OT (expressed in minutes) was carried out. In the non-obese group, the average LOS was 21.6 ± 18.0 days; among obese patients, it was 25.7 ± 15.4 days. The average OT in non-obese patients was 238.8 ± 76.2 min; in obese patients, it was 60.5 ± 64.2 min. Both parameters were slightly higher in the obese group; however, the association with obesity was not statistically significant.

### 3.3. Secondary End-Point: Pilot Study to Develop a Predictive Risk Score for Major Complications

Since obesity proved to be significantly associated only with DGF, BMI cannot be considered as the sole cause of worse outcomes in transplanted obese patients. Most importantly, BMI is not the correct index value for discriminating between eligible and non-eligible patients for kidney transplantation. Therefore, the BMI assessment must carefully evaluate each patient’s individual risk factors. This is performed to obtain a risk-predictive tool for major complications (delayed graft function and acute rejection) which can be used during the patients’ examination to determine their inclusion in the organ waiting list.

The univariate analysis evaluated the following possible risk factors: age, sex, BMI, hypertension, diabetes mellitus, dyslipidaemia, cardiovascular disease, and previous abdominal and/or urinary system surgery. However, hypertension, diabetes mellitus, dyslipidaemia, and cardiovascular diseases showed an OR < 1, and they did not prove to be risk factors for the onset of main complications. On the contrary, age, sex, BMI, previous abdominal and/or urinary system surgery, and the Charlson Comorbidity Score (adjusted for age and chronic kidney disease) were risk factors for the occurrence of significant complications (OR > 1). In the multivariate analysis, the effect of each variable from the univariate analysis was evaluated to determine its “pure” effect. Female sex (OR = 1.02, IC: 0.57–1.83, *p* = 0.939), age ≥ 57 years (OR = 1.10, IC: 0.64–1.90, *p* = 0.733), and previous abdominal and/or urinary system surgery (OR = 1.32, IC: 0.76–2.32 *p* = 0.326) showed an OR slightly above 1. Instead, the risk increased significantly concerning the BMI: in comparison to normal-weight individuals, overweight patients had a risk 98% higher (OR = 1.98, IC: 1.10–3.25, *p*-value 0.022), whereas the obese ones had a more than 2X probability of developing considerable complications (OR = 2.51, IC: 1.08–5.85, *p*-value = 0.033). Moreover, the Charlson Comorbidity Score was shown to be a predictor associated with the outcomes considered in this study: having a score ≥ 3, the risk of serious complications was double compared to patients with a score < 3 (OR = 2.06, IC: 1.06–3.97 *p* = 0.032) (Table 5). A nomogram, with the predictors of the multivariate model, was developed to calculate the score from which the a priori probability of notable complications was obtained (Figure 3). The nomogram included five variables referring to the patient’s risk factors, represented as separate and different lines. The sixth line of the normogram (the “Score”), shows the points attributed to each variable. The other two lines appear on the following representation: the seventh for the calculated probability of major complications, and the eighth for the total score of every patient, summing up all the “points” attributed to each variable. For instance, a female patient less than 57 years old, overweight, with no previous surgery and a Charlson Comorbidity Score ≥ 3 will have a total score equal to 0.5 (sex) + 0 (age) + 7.5 (BMI) + 0 (surgery) + 8 (Charlson) = 16 points. With a total score of 16, the probability of critical complications is about 0.42 (42%). The area under the curve (AUC) of the multivariate model was 0.6457 (95% IC: 0.57–0.72) (Figure 4). Therefore, the model can be considered fair for predicting the risk of major complications in both obese and non-obese individuals. The percentage of cases correctly identified by this model, retrospectively applied to the entire population, was 73.61%. Nevertheless, the model is highly specific (97.17%), but not too sensitive (7.89%), suggesting the need to consider other risk factors better to define the probability of major complications for prognostic purposes.

## 4. Discussion

Statistical analysis of both endpoints showed good statistical significance. Moreover, the two analyses are strictly related, with the second endpoint being the natural evolution of the first one. Kidney transplantation is the optimal treatment for patients with kidney failure. Every medical–surgical procedure is associated with risks and benefits whose ratio should be carefully evaluated to guarantee the best result. This must performed to avoid exposure to excessive risk in the case of non-suitable patients and/or in the case of potentially suitable patients only after targeted interventions. In obese patients, BMI plays a crucial role in evaluating the eligibility for renal transplantation. Considering the following issues—chronic organ shortage, the lack of guidelines, and international cut-offs in assessing obese candidates for kidney transplantation—and an existing ethically controversial approach, the identification of factors that could impact graft survival is fundamental to ensure the optimal use of the current poor organ resource and to avoid biased discrimination among obese candidates. The first endpoint of this study was that the BMI is not an adequate indicator for discriminating between suitable and non-suitable patients for kidney transplantation. The results obtained are partially consistent with those of the recent meta-analyses. The likelihood of access to being listed on a deceased donor waiting list and subsequently being transplanted is a challenge. Ladhani et al. (2020) showed that obesity was associated with a reduced likelihood of waitlisting, but not kidney transplantation once waitlisted. Moreover, women who were obese were 34% less likely to be listed than normal-weight women, compared to obese men, who were 14% less likely [25]. The international guidelines on kidney donor and recipient evaluation and perioperative care recommend that patients with a BMI > 30 kg/m^2^ should be advised to reduce weight before transplantation and that all prospective renal transplantation candidates must be evaluated for obesity using BMI or waist-to-hip circumference ratio at the time of listing and while on the waiting list. Also, even though they should not be excluded from transplantation, prospective obese kidney transplant candidates should be actively encouraged for weight loss interventions with either surgical or medical methods before transplantation. [26,27]. Nicoletto et al. [14] reported that a higher incidence of DGF was the only consequence of obesity. Lafranca et al. [15] showed no significant differences between obese and non-obese groups. Hill et al. [16] reported a correlation between obesity and DGF, while Sood et al. [17] highlighted how obese patients had an increased risk of DGF and acute rejection. Foucher et al. [18] pointed out a higher risk of graft loss, infections, DGF, and NODAT. Regarding early complications, by identifying DGF in obese patients as the only statistically significant complication, our study is mostly consistent with the results of the above-mentioned meta-analyses [14,15,16,17,18]. As for late complications, the results of the present study mainly deviate from those of the French study by Foucher et al. [18], since non-significant associations with obesity or neat differences between obese and non-obese groups were detected. The only statistically significant association was observed between obesity and DGF, suggesting the need for a different approach to better evaluate the transplant option in obese patients affected by chronic kidney disease. OR frequently are used to present the strength of the association between RF and outcomes in the clinical literature. The results from logistic regression are converted easily into OR because logistic regression estimates a parameter (the log odds), which is the natural logarithm of the OR. Logistic regression modelling allows for the estimates for an RF of interest to be adjusted for other RF—in our case, the BMI. Another promising feature is that it is easy to test the statistical strength of the association. Adding more independent explanatory variables to the model could have increased the OR of the variable of interest by dividing it by a smaller scaling factor. According to AIC, all models are approximations to reality, and reality should never have a low dimensionality. On the contrary, BIC tries to find the true model among the set of candidates. Due to the sample size, we have chosen to rely our results on the most reliable and reproducible test like the OR, including just the OR > 1 because of the known strength of the association with the outcome, instead of applying a more complex type of analysis. Therefore, the second endpoint of this work was analyzed. The prognostic risk score developed in this pilot study can be considered a promising model for assessing the probability of major complications and renal transplant failure. This study’s strength is the developed model’s fair performance in predicting the risk of major complications (95% IC: 0.57; 0.72) and its high sensitivity (97.17%). However, this study has some limitations. The tool obtained is not very sensitive (7.89%), and even though the sample size of this study is not that small, the development of a prognostic risk score further requires the evaluation of a broader population in a multicentre study. In the future, it will also be necessary to consider additional risk factors to obtain a higher-performance score.

## 5. Conclusions

In conclusion, this study, within the boundaries of its aforementioned limitations, points out that obese kidney transplant recipients seem to have an increased risk of DGF. In contrast, the risk of other early and late complications appears to be fairly equal to the one registered in the non-obese group. Therefore, obese patients can potentially receive similar benefits regarding transplant success and graft survival. Potential recipients should not be excluded a priori from transplantation merely based on BMI, but they should require the identification, assessment, and stratification of their risk factors. An easy-to-use score, like the one proposed in this paper, could be a valuable tool in identifying the need for interventions for patients to make them suitable and eligible for kidney transplants, reduce their peri-postoperative risk, improve their transplant outcomes, and guarantee their surgical safety. It could also provide a more ethically correct approach, as it evaluates patients based on properly validated and statistically significant individual risk factors.

## Figures and Tables

**Figure 1 life-14-00915-f001:**
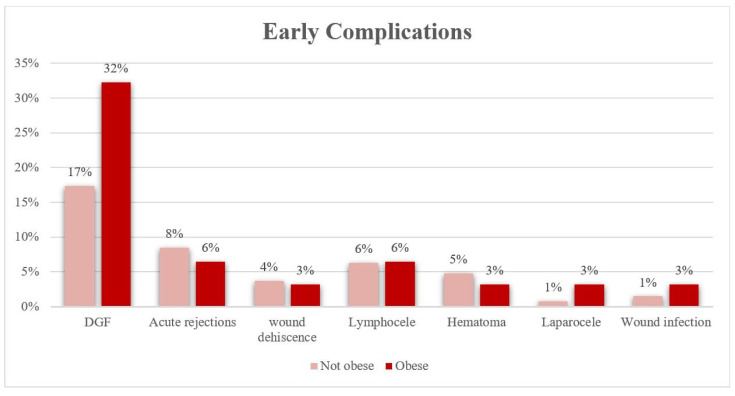
Distribution of early complications in obese and non-obese groups.

**Figure 2 life-14-00915-f002:**
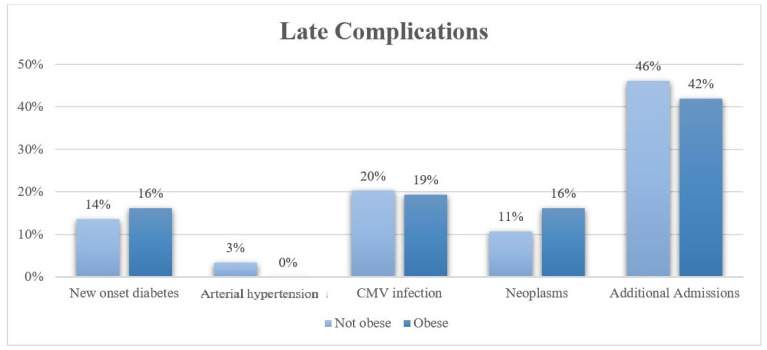
Distribution of late complications in obese and non-obese groups.

**Figure 3 life-14-00915-f003:**
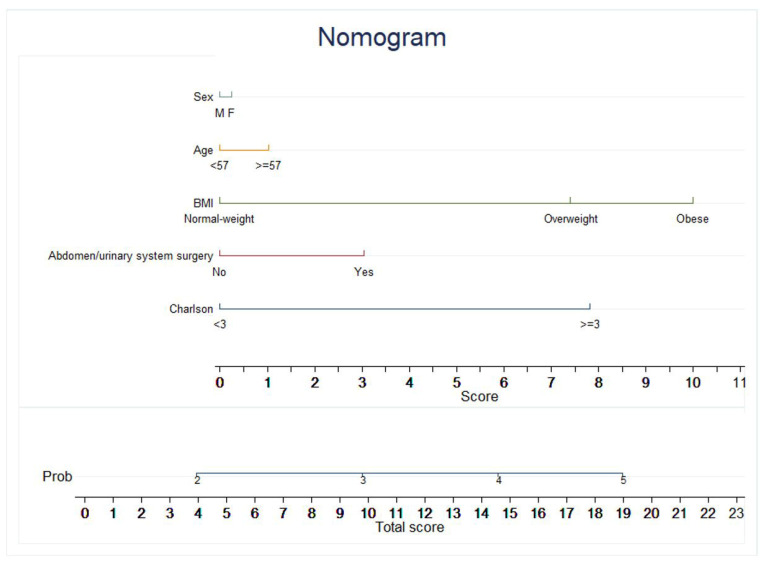
Nomogram to predict the probability of major complications.

**Figure 4 life-14-00915-f004:**
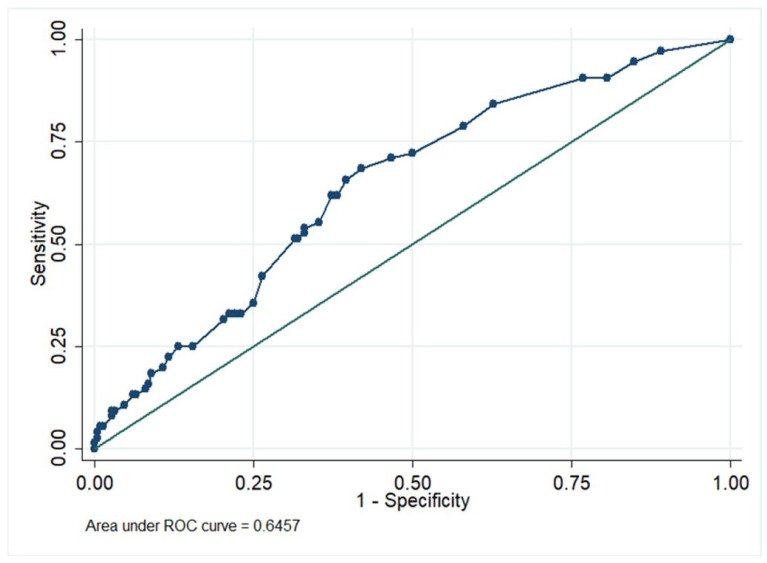
ROC curve of the major complications predictive model.

**Table 1 life-14-00915-t001:** Description of the cohort –qualitative variables.

		n	%
**Sex**	F	94	31.1%
	M	208	68.9%
**Year**	2014	39	12.9%
	2015	52	17.2%
	2016	35	11.6%
	2017	48	15.9%
	2018	38	12.6%
	2019	37	12.3%
	2020	35	11.6%
	2021	18	6.0%
**Patients**	Underweight	15	5,00%
	Normal	170	56.3%
	Overweight	101	33,40%
	Obesity (I)	26	8,60%
	Obesity (II)	5	1,70%
	Obesity (III)	0	0%
**Transplant**	Living	14	5%
	BHD	284	94%
	DCD	18	6%
**Charlson Comorbidity Score ≥ 3**		62	20.5%
**Charlson Comorbidity Score < 3**		240	79.5%
**Post-operative deaths**		5	1.7%
**Arterial Hypertension**		236	78.1%
**Displypidaemia**		35	11.6%
**Diabetes**		46	15.2%
**Cardiovascular diseases**		119	39.4%
**Abdominal/urinary system surgery**		186	61.6%
**Total**		302	100.0%

**Table 2 life-14-00915-t002:** Description of the cohort—quantitative variables.

	Average	Std. Dev.	Min	Max
**Age at transplantation**	53.2	11.7	19	76
**BMI (kg/m^2^)**	24.1	5.5	15.1	37.7
**Charlson Comorbidity Score, CCS**	1.3	1.5	0	7
**Length of stays, LOS (days)**	22	17.7	6	224
**Operative Time, OT (minutes)**	241	75.3	110	924

**Table 3 life-14-00915-t003:** Correlation between BMI and early complications.

	Non Obese		Obese		
	n	%	n	%	*p*-Value
**DGF**	47	17.3%	10	32.3%	0.044
**Acute rejection**	23	8.5%	2	6.5%	0.999
**Wound dehiscence**	10	3.7%	1	3.2%	0.999
**Lymphocele**	17	6.3%	2	6.5%	0.999
**Perirenal hematoma**	13	4.8%	1	3.2%	0.999
**Laparocele**	2	0.7%	1	3.2%	0.278
**Wound infection**	4	1.5%	1	3.2%	0.420

**Table 4 life-14-00915-t004:** Correlation between BMI and late complications.

	Non-Obese		Obese		
	n	%	n	%	*p*-Value
**New-Onset Diabetes**	37	13.7%	5	16.1%	0.706
**Arterial Hypertension**	9	3.3%	0	0.0%	0.605
**CMV infection**	55	20.3%	6	19.4%	0.902
**Neoplasms**	29	10.7%	5	16.1%	0.365
**Additional admissions**	125	46.1%	13	41.9%	0.657

**Table 5 life-14-00915-t005:** Association between identified risk factors and occurrence of major complications. Univariate and multivariate analysis.

		Univariate Model				Multivariate Model			
		OR	95% IC		*p*-Value	OR	95% IC		*p*-Value
**Sex**									
	**M**	Ref.				Ref.			
	**F**	1.03	0.59	1.796	0.908	1.02	0.57	1.83	0.939
**Age**						Ref.			
	**<57 years**	Ref.							
	**≥57 years**	1.28	0.77	2.16	0.344	1.10	0.64	1.90	0.733
**BMI**									
	**Normal weight**	Ref.				Ref.			
	**Overweight**	1.86	1.06	3.25	0.030	1.98	1.10	3.55	0.022
	**Obese**	2.49	1.10	5.62	0.028	2.51	1.08	5.85	0.033
**Previous surgery**									
	**No**	Ref.				Ref.			
	**Yes**	1.39	0.81	2.40	0.236	1.32	0.76	2.32	0.326
**Charlson Comorbidity Score**									
	**<3**	Ref.				Ref.			
	**≥3**	2.02	1.08	3.80	0.028	2.06	1.06	3.97	0.032

## Data Availability

The data presented in this study are available on request from the corresponding author due to privacy and local ethical restrictions.

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
