# Peer review of "Open Renal Transplantation in Obese Patients: A Correlation Study between BMI and Early and Late Complications with Implementation of a Prognostic Risk Score"

_life, 2024, doi:10.3390/life14070915_

Round 1

Reviewer 1 Report (Previous Reviewer 3)

Comments and Suggestions for Authors

Thank you for the opportunity to review again the article titled: Open Renal Transplantation In Obese Patients: A Correlation 2

Study Between BMI And Early And Late Complications With 3

Implementation Of A Prognostic Risk Score

The authors have responded and addressed each of my comments. They even defended some changes that were not made in the manuscript. However, I consider that the selection of only ORs > 1 is a mistake that must be considered. What happens with OR<1? The authors must consider this bias at least in the limitations of the study. On the other hand, AIC and BIC are the most objective tools we have to evaluate regression models.

Author Response

Reviewer 2 Report (New Reviewer)

Comments and Suggestions for Authors

Download the attachment

Comments on the Quality of English Language

Author Response

Reviewer 3 Report (New Reviewer)

Comments and Suggestions for Authors

The reviewed original work is an interesting introduction to issues related to the treatment options for various diseases in obese people, who constitute an increasing group of patients. The authors took into account various predictive factors in the analysis and conducted a retrospective study of their impact on the survival of people after kidney transplantation with comorbid obesity. The study result indicates the need to look for various markers to assess the success of therapy in this group of patients. Some of the results overlap with previous studies, but the final assessment has important clinical implications.

Editorial comments: The manuscript gives the impression that it is not the final version. At the beginning, there are several incomprehensible yellow underlines of the text, and there are shifts in the tables, which reduces the quality of the overall work. For example, there is no graphic diagram of the test procedure and no explanation of how to use the developed nomogram. Limitations should be separated in such a way that they can be easily found in the text. The literature is adequate, but quite old, which could indicate that the topic is not currently of interest, although the content proves otherwise. Perhaps it would be good to add a figure of the most important statistical analyses of correlations or the results of multivariate logistic analysis (in the main text, not in the appendices).

Comments on the Quality of English Language

no comments

Author Response

Reviewer 4 Report (New Reviewer)

Comments and Suggestions for Authors

Congratulations for the interesting paper. It is very much true that obese patients may develop a lot of problems and their number is constantly increasing. So, as you mentioned, it is of utmost importance to give a proper and ethical evaluation and treatment for these patients. As regard to easy risk score, I think we need further evaluation for proper validation. Meanwhile, individual score, taking in account comorbidities and all risk factors, remain the main and most adequate tool for these patients.

Author Response

This manuscript is a resubmission of an earlier submission. The following is a list of the peer review reports and author responses from that submission.

Round 1

Reviewer 1 Report

Comments and Suggestions for Authors

Thank you for giving me the opportunity to review this manuscript, which evaluated the impact of body mass index (BMI) on both perioperative and late complications after a kidney transplantation. The manuscript is generally well written. However, there are a few comments and shortcomings that preclude this to be published.

1.       End-stage renal disease (ESRD) - please consider the use of the new term “kidney failure” as per the latest KDIGO nomenclature recommendations.

2.       The author should include information on donor: % of cadaveric and living donor among obese group and non-obese groups, whether they are multi-organ donor (for cadaveric donors), their cold ischemic time, 2nd warm ischemic time, any intra-OT hypotension etc. These are all important factor that may affect the risk of DGF and thus should be accounted for.

3.       The author should also present the baseline characteristic as classified by obese vs non-obese groups, as it would be useful to appreciate whether there is any difference upfront between the two groups.

4.       The author should also mention that there may be selection or recall bias in the limitation section, given the nature of this being a retrospective study

Overall, while interesting, this topic has already been elegantly demonstrated by several group (Scheuermann et al, BMC Nephrol. 2022; Bellini et al, Front Endocrinol. 2023). Thus, this paper unfortunately adds little to the current literature.

Reviewer 2 Report

Comments and Suggestions for Authors

  1. The limitations of this study include a small sample size and the collection of all samples from a single center, which may not represent the broader population and carries a risk of bias.
  2. The AUC of 0.6457 suggests that the model may have low accuracy.
  3. In the introduction, lines 70-77, the authors mention that robotic surgery decreases the incidence of surgical site infections after kidney transplantation. However, it lacks connection with the context and the rest of the manuscript.
  4. In lines 154-155, the text states, “Among the 302 patients, 31 were obese (10.0%), of whom 5 were class II obese – and 101 were overweight (33.4%).” Please include the BMI ranges for the different degrees of obesity here. Additionally, It`s better list the number and percentage of patients of all obesity degrees in Table 1.
  5. In Table 1, the author lists the total number and percentage of patients following all categories, such as gender, year, and degrees of obesity, which can be confusing. It would be clearer to include the total number and percentage of patients after each category.
  6. The manuscript contains minor errors, such as the incorrect formatting of "kg/m2" where "2" should be superscript. 
  7. It`s better ensure that Figures 1 and 2 are of the same size.

Reviewer 3 Report

Comments and Suggestions for Authors

Thank you very much for the opportunity to review the article titled: “Open Renal Transplantation In Obese Patients: A Correlation Study Between BMI And Early/Late Complications With Implementation Of A Prognostic Risk Score”, by Marzorati and colleagues. This work presents an interesting proposal to identify whether BMI is associated with pre- and postoperative complications of renal transplantation. Additionally, in this study, a nomogram was developed in which scores for risk factors were tested. I have some comments related to the presentation and analysis of the results of this work.

1.    It is important to describe and include in the analyses whether the transplants were from a living or deceased donor.

2.    On lines 121 and 122, the authors describe the following: “The age cut-off was set at ≥ 57 years (similar to the literature which suggests being set at ≥ 60 years in surgical predictive scores)”. I consider it important to test other cut-off points and not solely rely on what is reported in the literature. Interesting findings could emerge in this regard.

3.    On lines 140-141, the authors specify the following: “Variables with an odds ratio (OR) of >1 in the univariate analysis were included in the multivariate logistic regression analysis. All variables that remained significant in the multivariate model were used to build a scoring system.” I consider this an error, as there can be variables with an OR less than 1 that are significant, or conversely, variables with a value greater than 1 that do not contribute to the model. The selection of variables should be based on stricter criteria such as AIC and BIC, or at least take into account confidence intervals.

4.    Add confidence intervals for reporting ORs and not just report the crude OR in the text.

5.    I consider the inclusion of confidence intervals for all tables where comparisons are made enriching for the study.

6.    On lines 208 to 210, the authors describe the following: “Instead, the risk increased significantly in relation to the BMI: in comparison to normal-weight individuals, overweight patients had a risk 98% higher (OR=1.98, p-value 0.022),”. This is an incorrect interpretation for an OR. The OR compares odds and not direct probabilities (risks), and although odds and risks can provide similar information when the incidence of the event is low, they are not the same and should not be used interchangeably. Please change the "risk" OR interpretation in the study. 

7.    I consider that the nomogram could improve if various models with different variables are tested, not only including those that are significant. As reported and described in the results of the logistic regressions, it is not possible to assess whether the addition or elimination of variables affects the models or if interactions are present, etc. Rethink the model and test various models; it is suggested to use AIC and BIC.

8.    The authors do not describe any aspects related to the assumptions and validation of the multivariable logistic regression. Include this.

9.    Include confidence intervals for Figure 4.

10. Figure 1 shows comparisons of some characteristics between obese and non-obese patients; it is recommended to perform a hypothesis test.

11. It is not clear what happened with the 101 overweight patients.

12. The researchers considered reanalyzing the data by grouping obese and overweight patients together.

13. Include in Table 1 the number of patients without obesity.